# Phenolic Acid Profiling of *Lactarius hatsudake* Extracts, Anti-Cancer Function and Its Molecular Mechanisms

**DOI:** 10.3390/foods11131839

**Published:** 2022-06-22

**Authors:** Qiao Yang, Xiaoyi Zhang, Huini Qin, Feijun Luo, Jiali Ren

**Affiliations:** 1Hunan Key Laboratory of Forestry Edible Sources Safety and Processing, Changsha 410004, China; yq15074457162@163.com (Q.Y.); qunqundexiao1@gmail.com (X.Z.); huini_q@163.com (H.Q.); t20121480@csuft.edu.cn (F.L.); 2College of Food Science and Engineering, Central South University of Forestry and Technology, Changsha 410004, China

**Keywords:** *Lactarius hatsudake*, phenolic acids, anti-cancer, molecular mechanism

## Abstract

Cancer is still the leading cause of death across the world, and there is a lack of efficient therapies. *Lactarius hatsudake* is a mushroom with a food and medicine homology that contains numerous biologically active substances. This study aimed to investigate the composition of extracts from *Lactarius hatsudake* (*L. hatsudake*) and their anti-cancer function and molecular mechanisms. Our results showed that the total phenolic content of *L. hatsudake* extracts was 139.46 ± 5.42 mg/g. The following six phenolic compounds were identified from *L. hatsudake* extracts by HPLC and UPLC-QTOF/MS: gallic acid, pyrogallol, chlorogenic acid, ferulic acid, myricetin, and cinnamic acid. Colorectal cancer cell HCT116 and hepatic cancer cell HepG2 were used to evaluate the anti-cancer function of the *L. hatsudake* extracts. Compared with HepG2 cells, the *L. hatsudake* extracts showed stronger anti-cancer activity against HCT116 cells and these were used to study molecular mechanisms. The results indicated that the *L. hatsudake* extracts could arrest the cancer cell cycle and inhibit cancer cell proliferation, which may be mediated by the MAPK/NFκB/AP-1 signalling pathway; the *L. hatsudake* extracts also promoted cancer cell apoptosis through a mitochondrial-dependent pathway. Taken together, these findings demonstrate that *L. hatsudake* ethanol extracts contain six main phenolics and illustrate the remarkable potentiality of *L. hatsudake* as a source of natural phenolics for cancer prevention and as an adjuvant in the treatment of functional foods.

## 1. Introduction

Cancer is still the main cause of death across world. According to the estimation of the World Health Organization (WHO) [1], there were about 19.29 million new cancer cases and 9.96 million deaths in 2020 worldwide. Cancer is a kind of sporadic disease caused by abnormal mechanisms of cell proliferation and apoptosis, which are induced by a variety of factors such as environmental, pathological, and biological mechanisms. The current anti-cancer drugs in clinical therapy show side effects and complications compared with natural bioactive products [2]. Therefore, developing natural compounds to reduce the major cancer risk factors and prevent carcinogenesis becomes necessary. Recently, mushroom-derived secondary metabolites have attracted greater attention in chemoprevention. The compounds can kill or inhibit tumor cells as cytotoxic anti-tumor drugs [3]. For example, 200 and 300 mg/kg of the *Ganoderma lucidum* polysaccharide decreased AOM/DSS-induced tumorigenesis in a dose-dependent manner [4]. Their anti-cancer activities are closely related to carcinogenesis-related signal pathways such as PI3K/Akt pathway, NFκB pathway, MAPK pathway, etc. Further investigations of the anti-cancer effects of mushrooms or their active compounds has been conducted, and indicates its great potential market value in the future [5,6].

*Lactarius hatsudake* (*L. hatsudake*), a great wild endemic fungus belonging to the genus *Lactarius* family *Russulaceae* with a food and medicine homology, is widely distributed in Asia, Europe, and North America [7]. It is known for its abundance of major nutrients and its high content of bioactive compounds, e.g., polyphenols, polysaccharides, and ergosterols [8]. Several studies have indicated that mushroom polysaccharides have anti-tumor activities, ergosterols can promote calcium absorption and polyphenols have an antioxidation effect. *Lactarius* extracts showed the greatest anti-glioma potential compared with other mushrooms, but there are very few studies focusing on elucidating the key ingredients of its anti-tumor activity. Polyphenols are comprised of a variety of bioactive compounds that are usually divided into several classes such as flavonoids, phenolic acids, and lignans [9]. The phenolic acids can reduce the adhesion of various microbial cells, which ultimately leads to reduced causes of diseases [10]. It has been reported that polyphenols composed of hydroxycinnamic acid derivatives, flavan-3-ols, and flavonols could inhibit the progress of glioblastoma, colon cancer, and lung carcinoma in a significant dose-dependent manner [5,11]. Other studies showed that geraniin and isocorilagin exhibited higher cytotoxicity against MCF-7 with an IC50 of 13.2 and 80.9 μg/mL [12]. Those investigations suggest that extracts including phenolic from *L. hatsudake* may be the important compounds exerting anti-cancer effect.

In this study, the aim was to evaluate the anti-tumor activity of *L. hatsudake* extracts. *L. hatsudake* extracts were obtained with 60% ethanol under ultrasonic conditions, and then purified with AB-8 macroporous resin. Six phenolics were identified by using both HPLC and UPLC-QTOF/MS [13]. Their anti-cancer activities were determined using in vitro assay with HCT116 cells and HepG2 cells. The effects of *L. hatsudake* extracts on cell proliferation and apoptosis were assessed and the anti-cancer mechanisms were explored, as shown in Figure 1.

## 2. Materials and Methods

### 2.1. Chemicals

The fresh *L. hatsudake* fruit bodies were purchased from Hunan, Changsha Province, China. The standards of myricetin, cinnamic acid, and pyrogallol were purchased from Shanghai Aladdin Biochemical Technology Co., Ltd. (Shanghai, China). Caffeic acid, chlorogenic acid, ferulic acid, and gallic acid were purchased from Sigma-Aldrich (Shanghai, China) Trading Co., Ltd. Acetonitrile, methanol, and formic acid were of HPLC grade and purchased from Sigma-Aldrich Trading Co., Ltd. HCT116 and HepG2 cells were obtained from the Cell Culture Center of Shanghai Institute for Biological Sciences (Shanghai, China). Additionally, 0.25% trypsin-EDTA, fetal bovine serum (PBS), 1640 medium, penicillin-streptomycin, and trypsin were purchased from Gibco (Waltham, MA, USA). BSA was purchased from Beijing Solaibao Biotech Water Co., Ltd. (Beijing, China). MTS was purchased from Millipore and SYBR Green was purchased from the Beijing Quansi Gold Biotechnology Co., Ltd. (Beijing, China).

Other chemicals not mentioned above were of HPLC or analytical grade. Aqueous solutions were prepared with deionized water (18.2 MΩ·cm; Simplicity 185, Millipore Corp, Billerica, MA, USA).

### 2.2. Extraction and Purification

The protocols of this study are similar to those of reference [14], with some modifications. The clean fresh *L. hatsudake* fruit bodies were dried under freeze-drying, and then processed through a 100 mesh sieve after being ground to a powder. The powders were extracted using 60% ethanol in the ratio of 1:25 (*w*/*v*) and ultrasonic assisted for 80 min at 30 °C. The mixture was centrifuged at 3000 rpm for 15 min. The residues were re-extracted twice under the same conditions and all the ethanolic extracts were combined. The organic solvents in the extracts were removed with a rotary evaporator (N-1300V-W, Tokyo Rikakikai Co., Ltd. (Tokyo, Japan)) and then the residues were vacuum freeze-dried.

The *L. hatsudake* extracts were purified with AB-8 macroporous resin and the obtained compounds were condensed with a rotary evaporator and dried with a vacuum freeze dryer (Ytlg-12A, Shanghai Yetuo Co., Ltd. (Shanghai, China)).

### 2.3. Total Phenolic Content

The total phenolic contents with *L. hatsudake* ethanolic extracts was measured by means of the Folin–Ciocalteu assay [15]. Gallic acid was used to calculate the standard curve. Estimation of the total phenolic acids was carried out in six duplicates (*n* = 6). The result was expressed as mg of gallic acid per gram of the extracts.

### 2.4. Analysis of Phenolic Profiles

The composition of *L. hatsudake* extracts was analyzed using both the HPLC and UPLC-QTOF/MS methods described by Palacios [16]. An LC-20A HPLC (Shimadzu, Japan) equipped with a UV-vis detector was used for profiling. Separation was achieved on an ODS-C_18_ column (5 µm, 250 mm, 4.6 mm i.d.) at 30 °C by using the specific solvent system of 0.1% acetic acid in water (solvent A) and acetonitrile (solvent B) under the optimum gradient conditions (0–2 min, 10% B; 2–10 min, 20% B; 10–30 min, 80% B; 30–40 min, 100% B, 40–43 min, 8% B). The standard phenolic compounds (gallic acid, pyrogallol, chlorogenic acid, caffeic acid, ferulic acid, myricetin, and cinnamic acid) were used. A Waters ACQUITY UPLC IClass/Xevo in line with a Waters Xevo G2 Q-TOF mass spectrometer (Milford, MA, USA) was also used to detect the *L. hatsudake* extracts. Chromatographic separation was performed with a Luna C_18_ (2) column (250 mm × 4.6 mm, 5 μm; Phenomenex (Aschaffenburg, Germany)), with the temperature kept at 40 °C. Electrospray ionization (ESI) was performed in the negative mode (capillary temperature: 350 °C; capillary voltage: −5 kV; spray voltage: −4 kV).

### 2.5. Cell Culture and Polyphenol Treatments

The HCT116 cells were cultured in an RPMI-1640 medium including 10% (*v*/*v*) fetal bovine serum (FBS) and 1% penicillin/streptomycin (100 U/mL penicillin and 100 mg/mL streptomycin). *L. hatsudake* extracts were dissolved in ethanol and then diluted with ultrapure water to achieve the desired ratio of alcohol. The final concentration of *L. hatsudake* extracts was 10 mg/mL and they were stored in a 4 °C refrigerator. The stock solution was diluted with the basal medium to prepare the required working fluid concentration. The final concentration of ethanol was less than 0.1%, which did not affect cell viability.

### 2.6. Cell Viability Assay

The cytotoxicity of *L. hatsudake* extracts was detected using an MTS assay. Briefly, HCT116 cells were seeded in 96-well culture plates at a density of 4 × 10^4^ cells per well at 37 °C for 8–10 h. Next, cells were treated with different concentrations (0, 6.25, 12.5, 25, 50, 100, 200 μg/mL) of *L. hatsudake* extracts in culture medium for 24 h. Subsequently, 20 μL MTS solution (dissolved in PBS) was added to the well of plates, and the plates were incubated at 37 °C for 4 h. The absorbance was measured at 490 nm with a microplate reader to assess the optical density (OD), from which cell counts were determined. The inhibition rate of *L. hatsudake* extracts was calculated as follows:Inhibition rate (%) = (1-experimental group OD/control group OD) × 100%.

### 2.7. Hoechst 33258 Fluorescent Staining

HCT116 was digested into individual cells and then prepared into cell-like slices. When the cells adhered to the wall, *L. hatsudake* extracts of different concentrations (0, 25, 50, 100 µg/mL) were added. Afterwards, cells were treated at 4 °C for 5 min by using cell fixative (methanol: ice acetic acid = 3:1). After the PBS was gently cleaned, a solution of 5 g/mL Hochest 33258 was added for 10 min, then washed three times with PBS, each time for 5 min. After drying, the cellulate side of the slide was covered downward on the slide, and the slide was closed with transparent nail polish and observed under a fluorescence microscope.

### 2.8. SDS–PAGE and Western Blot Analysis

HCT116 cells were cultured at a density of 1 × 10^6^ cells/mL in a Petri dish with a diameter of 10 cm. After 12 h, fresh medium was added at 8 mL per dish and *L. hatsudake* extracts were added for a final concentration of 0, 25, 50, and 100 ug/mL. After continued culturing for 16–18 h, the culture solution was discarded, washed twice with ice-cold PBS, and the PBS in the Petri dish was aspirated with a pipette. A total of 300 μL of the RIPA lysate containing phosphatase and protease inhibitors was added to each well. The cells were scraped, and the lysate was pipetted into a 1.5 mL centrifuge tube and centrifuged at 13,000 rpm for 15 min at 4 °C. The supernatant was aspirated to obtain the total protein. The protein concentration was determined using a BCA kit to determine the amount of the sample. After the 5 × SDS buffer (10% SDS, 50% glycerol, Tris-HCl pH6.8 125 mM) was added to the above supernatant, it was immediately heated at 95 °C for 5 min. The equivalent protein sample was subjected to SDS-PAGE and electrophoresis transfer to the PVDF membranes semidry transfer device (Bio-rad, Hercules, CA, USA). Nonspecific binding sites in PVDF membranes were sealed with 5% bovine serum albumin in TBST (20 mM Tris, 166 mM NaCl, and 0.05% Tween 20, pH 7.5) for 1 h. The PVDF membrane with an appropriate primary antibody was stored at 4 °C overnight, and then incubated along with their respective horseradish-peroxidase conjugated secondary antibody at room temperature. Finally, the PVDF membrane was washed three times with TBST. It was detected using the chemiluminescent substrate (Thermo Fisher, Waltham, MA, USA) and visualized with the Molecular Image chemdoc XRS system (Bio-rad, Hercules, CA, USA).

### 2.9. Reverse Transcription-Quantitative PCR (RT-qPCR) Analysis

HCT116 cells were seeded at a density of 1 × 10^6^/well in a 6-well culture dish and placed in a carbon dioxide incubator for 24 h. Cells were treated with *L. hatsudake* extracts (0, 25, 50, and 100 μg/mL) for 12 h. Extraction of total RNA from cells was performed using the Transzol-Up kit according to the manufacturer’s instructions. The mass, purity, and concentration of the RNA samples were analyzed using a Nanodrop ultra-differential photometer instrument. A total of 2 μg of RNA was added to a reverse transcription system of 20 μL, and cDNA synthesis was performed using a high-efficiency cDNA reverse transcription kit. qPCR reactions were observed using SYBR Green I (Trans Gen Biotech Co., Ltd.; Beijing, China) in accordance with the manufacturer’s protocol. The amplification conditions were as follows: 94 °C, initial denaturation for 3 min; 94 °C, denaturation for 30 s; 60 °C, annealing for 40 s; and 72 °C, extension for 1 min, 40 cycles. Relative expression levels of the target genes were calculated by using 2−△△Ct(RQ)method. For PCR primers, see Table 1.

### 2.10. Cytometry Analysis

HCT116 cells were treated with different concentrations of *L. hatsudake* extracts for 48 h. Cells were washed twice with D-Hanks at 4 °C and the supernatant was discarded. They were fixed for 12 h with 70% ethanol in ice-cold PBS at −20 °C, centrifuge and ethanol was discarded, they were washed twice with PBS, and 100 µL was added. The FITC annexin V apoptosis detection kit (BD Biosciences, San Jose, CA, USA) was used for DNA extraction. A DNA extraction buffer reaction occurred at room temperature for 30 min. Then, 1 × PBS was used for washing the cells, the cells were resuspended using 100 µL 1 × PBS, 10 µL 10 mg/mL RNase and 10 µL 1 mg/mL PI solution were added, and they were incubated at room temperature in the dark for 30 min. An FACS Caliber Flow Cytometer (BD Biosciences, San Jose, CA, USA) was used to detect the cell cycle.

### 2.11. Apoptosis Detection

The HCT116 cells were up to 90% full, monolayer adherent cells were digested with 0.05% trypsin without EDTA, and the cell concentration was diluted to 5 × 10^5^ cells/mL with medium and inoculated into a six-well plate in an amount of 1 mL per well. Then, they were incubated overnight, the medium was discarded, and cells were washed twice with PBS, after which 1 mL of fresh medium was added alongside *L. hatsudake* extracts to a final concentration of 0, 25 μmol/L, 50 μmol/L, and 100 μmol. After 12 h, the HCT116 cells were digested and collected, and then centrifuged at 1000 rpm for 4 min at 5 °C, the supernatant medium was removed, resuspended in PBS, centrifuged again, and the supernatant was repeated once, and placed on a filter paper. After about 1 min, the PBS was removed. A total of 400 μL of the 1 × Annexin Binding buffer was added and cells were gently resuspended. Then, 5 μL of Annexin V-FTTC and 5 μL of PI were added to each of the groups, which were incubated at room temperature for 10–15 min in the dark, and checked on the machine within 1 h.

### 2.12. Luciferase Reporter Activity Assay

HCT116 cells in the logarithmic growth phase were obtained, the culture solution was drained, cells were washed twice with D-Hanks buffer, the residual liquid was aspirated, and the cells were digested by adding the appropriate amount of trypsin. The flask was removed and to stop digestion 1 ml of the 1640 medium was added and gently pipetted to make a cell suspension. After counting using the cytometer, the medium was diluted to a concentration of 4 × 10^6^ /mL. The cell suspension was inoculated into a 24-well plate at 500 μL per well. After incubation overnight, the cells were grown by approximately 70–80%, the medium was aspirated, cells were washed twice with ice-cold PBS, and the remaining PBS was blotted in the six-well plate with a pipette. With serum-free medium, the cells were washed, which was then replaced with 500 μL per well of fresh serum-free medium. A total of 1.5 μL Lipo2000 was added to each well, and 1.0 μg of AP-1 or NF-κB plasmid was added, mixed gently, and absorbed for 3 h. After removing the medium and washing once with PBS, the normal medium was added, incubated for 5 h at 37 °C, 5% CO_2_ was added and *L. hatsudake* extracts at 25 μg/mL, 50 μg/mL and 100 μg/mL were added. Culturing continued for 12 h, the culture medium was discarded, 200 μL of the cell lysate was added, then cells were incubated at room temperature for 5 min, and after lysis, transferred to a 1.5 mL EP tube, centrifuged at 10,000 rpm for 5 min, and the supernatant was aspirated. A total of 50 μL of the treated supernatant was added to the assay plate for 3 replicates at each concentration. A total of 80 μL of luciferase was added before the assay and detected by the machine.

### 2.13. Statistical Analysis

The phenolic compound content of *L. hatsudake* extracts was expressed as the mean ± SD (standard deviation) of the six replicates. Statistical analysis was conducted using SPSS19 software (SPSS, Inc., Chicago, IL, USA) with a one-way analysis of variance (ANOVA), and a value of *p* < 0.05 was taken as the level of significance.

## 3. Results and Discussion

### 3.1. The Contents and Composition of Total Phenols

The total phenolic content obtained from the crude extract and the purified extract from the *L. hatsudake* samples (*n* = 6) was found. The total phenolic content of the crude extract was 5.05 ± 0.15 mg/g as equivalents of gallic acid (Appendix A) under the optimal conditions which were as follows: ethanol concentration 60%, solid–liquid ratio 1:25 (*w*/*v*), and extraction time of 80 min, as shown in Appendix A. After purification by AB-8 macroporous resin, the phenolic content increased to 139.46 ± 5.42 mg/g as equivalents of gallic acid. The total phenolic content in *L. hatsudake* samples was calculated from the crude extract and 0.72 ± 0.02 mg/g was obtained.

The phenolic profiles of the purified extract in *L. hatsudake* were analyzed by both HPLC and UPLC-QTOF/MS. As shown in Figure 2, the extracts from *L. hatsudake* mainly contained gallic acid, pyrogallol, chlorogenic acid, ferulic acid, myricetin, and cinnamic acid compared with standard substances [17]. Interestingly, all phenolic compounds from *L. hatsudake* extracts belonged to phenolic acids from HPLC detection, shown in Figure 2B. However, the sample and standard substances have some deviations in retention time due to bad parallelism with HPLC. To further verify whether the results are reliable and to explore the specific composition, the results were observed using UPLC-QTOF/MS with standard substances. According to Table 2, gallic acid, pyrogallol, chlorogenic acid, ferulic acid, myricetin, and cinnamic acid were detected at similar rates as for the HPLC results, and the results are also consistent with previous articles [18,19,20]. Besides that, there are another seven unknown substances which appeared that required further exploration. Previous studies have shown that phenolic acids exhibit obvious bioactivities to reduce the occurrence of disease [21]. Some groups found that chlorogenic acid decreased colon cancer cell proliferation with major colonic microbial metabolites as a result of anti-proliferative effects, S-phase cell-cycle arrest and apoptosis in human colon cancer Caco-2 cells [22]. Another study found that gallic acids could induce apoptosis in HL60 cells due to the production of H_2_O_2_ [23]. The results indicated that the phenolic content in *L. hatsudake* extracts promoted the potential for anti-cancer activities.

### 3.2. L. hatsudake Extracts Inhibited Cancer Cell Proliferation

The notion of chemoprevention by natural bioactivities compounds has received much attention in the past decade, especially in anti-cancer activity with mushroom extracts [24]. To detect the mechanism of the anti-cancer function of *L. hatsudake* extracts, colorectal cancer HCT116 and hepatic cancer cell HepG2 were tested in vitro. The two kinds of cells were added at various concentrations (0, 6.25, 12.5, 25, 50, 100, and 200 µg/mL) of *L. hatsudake* extracts for 24 h, and cell viability was analyzed using the MTS assay. Under an optical microscope, the morphology of two cancer cells ruptured and died, and the number of cancer cells decreased (Figure 3A,B). The MTS experiment results showed that *L. hatsudake* extracts inhibited the cell proliferation of HCT116 with 72% cell viability and HepG2 with 75% cell viability at 200 µg/mL. With the increase in the *L. hatsudake* extracts concentration, cell viability decreased. Compared with the phenolic extracted from Sorghum, Bran had a significant inhibitory effect at 1.5 mg/mL, and *L. hatsudake* extracts inhibited cell proliferation at a quantity of 25 µg/mL [11] (Figure 3C,D). Additionally, the anti-cancer effect of *L. hatsudake* extracts is a little more sensitive to HCT116 cells and we used the cell line to study further experiments. Our results suggest that *L. hatsudake* extracts can prevent the proliferation of cancer cells.

### 3.3. L. hatsudake Extracts Arrest Cell Cycle of Cancer Cells

To better understand the bioactivity of *L. hatsudake* extracts, a study on the mechanism of its antiproliferative activity against HCT116 cells is currently underway. One of the mechanisms of the prevention of cancer is regulating the cell cycle, which can arrest the cell cycle at the G0/G1, S, or G2/M phase. To assess the effect of *L. hatsudake* extracts on the cell cycle, colorectal cancer HCT116 cells were added to 0, 25, 50, and 100 µg/mL *L. hatsudake* extracts for 24 h. The cytometry results showed that *L. hatsudake* extract treatments caused the percentage of the G1-phase to increase from 34.2% to 41.7%, 45.1%, and 49.0%, respectively. On the contrary, the percentages of the S-phase and G2/M-phase also reduced in a dosage-dependent manner (Figure 4A). The data suggest that *L. hatsudake* extracts can increase the percentage of resting cancer cells and decrease the percentage of mitotic cancer cells, which cause cell cycle arrest. The most important feature of cancer cells is that the proliferation of cells is out of control, and *L. hatsudake* extracts can inhibit the proliferation of cancer cells, meaning that it has an anti-cancer effect.

### 3.4. L. hatsudake Extracts Decreased CDKIs and Cyclin D1 Expressions

The cell cycle is regulated by the Cyclin/CDK/CDKI complex system. Our data indicated that *L. hatsudake* extracts increased the CDKI mRNA expression levels of p18, p21, and p53 in a dosage-dependent manner (Figure 3B). The Western blotting analysis further proved that *L. hatsudake* extracts reduced cyclin D1 protein expression levels and increased p21 and p53 protein expression (Figure 4C). p18 can bind with CDK4 and CDK6 and inhibits their activities [25], and p21 can bind CDK2, CDK4, and CDK6 and reduce their activities, which promote down-regulated activities and cell cycle arrest [26]. Wild-type p53 is a tumor suppressor gene, and up-regulated p53 expression can promote p21 expression, which inhibits CDKs and results in cell cycle arrest. Meanwhile, up-regulated p53 expression can promote cancer cell apoptosis [27]. *L. hatsudake* extracts could increase expressions of p18, p21, and p53, suggesting that *L. hatsudake* extracts can inhibit cancer cell proliferation by increasing the expression of CDKIs and inhibiting CDK activities. Our data showed that *L. hatsudake* extracts downregulated cyclin D1 mRNA expression due to RT-qPCR assay in HCT116 cancer cells, and the decreased expression of cyclin D1 reduced the binds to CDK4 and CDK6, which inhibited RB in releasing transcription factor E2F and reduced proliferation-related gene expressions [28]. Taken together, the results demonstrate that *L. hatsudake* extracts may prevent the cell cycle by decreasing cyclin D1 expression and increasing CDKI expressions.

### 3.5. L. hatsudake Extracts Inhibited Transcript Activities of NFκB and AP-1

Transcription factor activator protein 1 (AP-1) is a dimeric complex and its members include Jun, Fos, ATF, and MAF. AP-1 activation is closely related to carcinogenesis and is mostly activated in cancer. In the promoter of cyclin D1, AP-1 binding sites exist and AP-1 regulates the expression of cyclin D1. In this study, *L. hatsudake* extracts inhibited the transcript activities of AP-1, as found in the luciferase reporter gene analysis, in a dosage-dependent manner (Figure 5A) and *L. hatsudake* extracts decreased the protein expression of cyclin D1 (Figure 4C), suggesting that *L. hatsudake* extracts may decrease cyclin D1 expression by inhibiting AP-1 activation. Transcript factor NFκB has long been considered as a prototypical proinflammatory signaling pathway, largely based on the activation of NFκB by proinflammatory cytokines such as interleukin 1 (IL-1) and tumor necrosis factor α (TNFα). NFκB activation also participates in carcinogenesis. In this study, *L. hatsudake* extracts inhibited the transcript activities of NFκB, as found in the luciferase reporter gene analysis, in a dosage-dependent manner (Figure 5B). This indicates that *L. hatsudake* extracts may decrease cyclin D1 expression by inhibiting NFκB activation. Taken together, *L. hatsudake* extracts may downregulate the activities of NFκB and AP-1 and inhibit cyclin D1, which results in the cell cycle arrest of cancer cells.

### 3.6. L. hatsudake Extracts Inhibited MAPK Signal Pathway

MAPKs are always activated in cancer cells and uncontrolled cancer cell proliferation is closely related to MAPK activation. Our data indicated that *L. hatsudake* extracts decreased the phosphorylated proteins of ERK1/2 and JNK, as observed in the Western blotting analysis (Figure 5C), which means *L. hatsudake* extracts can prevent MAPK activation. Interestingly, ERK1/2 can cause the phosphorylation of c-Fos (one member of AP-1) and JNK can cause the phosphorylation of c-Jun (another member of AP-1) [29], which suggests that *L. hatsudake* extracts inhibit the transcript activation of AP-1 by decreasing MAPK activities. ERK1/2 also can cause NFκB activation [30]. Our results showed that *L. hatsudake* extracts inhibit the transcript activation of NFκB, proving that *L. hatsudake* extracts inhibit the transcript activation of NFκB by decreasing MAPK activities. Taken together, our data indicate that *L. hatsudake* extracts block MAPK activation and downregulate the activities of NFκB and AP-1, which further inhibit cyclin D1 and the proliferation of cancer cells.

### 3.7. Pro-Apoptosis Effect and Mechanism of L. hatsudake Extracts

Apoptosis is a physiological and crucial process that is regarded as the preferred way to eliminate cancer cells [31]. Colorectal cancer cell HCT116 was treated with *L. hatsudake* extracts for 24 h, cells were fixed with a mixture solution of methanol: glacial acetic acid. Then, HCT116 cells were stained using Hoechst33258 fluorescent dye and the cell morphology was observed with a fluorescent microscope. We found that *L. hatsudake* extracts treatment resulted in nuclear condensation and nuclear fragmentation, and apoptotic bodies appeared (Figure 6A), as indicated by the arrow.

To explore the molecular mechanism of the promoting apoptosis of *L. hatsudake* extracts, various concentrations of *L. hatsudake* extracts were added to HCT116 cells. The Western blotting analysis found that *L. hatsudake* extracts increased cleaved Caspase-3 protein and promoted cancer cell apoptosis. *L. hatsudake* extracts also increased the expression of promoting apoptosis proteins Bax in a dosage-dependent manner; on the contrary, anti-apoptotic protein Bcl-2 expression was reduced. All of these proteins involved in the downstream events of mitochondria and Bcl-2, depended on the mitochondrial apoptosis pathway [32] (Figure 6B). These findings suggests that *L. hatsudake* extracts may promote cancer cell apoptosis by altering Caspase-3 and the ratio of Bcl-2/Bax [33]. Meanwhile, *L. hatsudake* extracts activate p38 and may also promote the apoptosis of cancer cell HCT116 (Figure 6). Aside from altering apoptosis gene expression, DNA damage is also an important factor in HCT116-cell apoptosis, which is an important molecular target in the killing of tumor cells [34,35]. Importantly, the decreased phosphorylated proteins of ERK1/2 and JNK, were found to be downstream target kinases of DNA damage-induced apoptosis [36]. These data suggest that *L. hatsudake* extracts can promote HCT116-cell apoptosis via the mitochondria-dependent pathway.

## 4. Conclusions

In conclusion, the extracts of *L. hatsudake* mainly consisted of six phenolic compounds, which we found by conducting an HPLC/HPLC-MS analysis. *L. hatsudake* extracts can inhibit the MAPK pathway and the transcript activities of NFκB and AP-1, which modulate the expressions of cyclin D1. The down-regulated expression of cyclin D1 and up-regulated expression of CDKIs can reduce CDKs’ activities and cause cell cycle arrest, which prevents cancer cell proliferation. Meanwhile, *L. hatsudake* extracts can induce the apoptosis of cancer cells. *L. hatsudake* extracts can activate apoptotic effector protein kinase Caspase-3, change the ratio of Bcl-2/Bax and cause mitochondrial damage, which means *L. hatsudake* extracts exert an apoptotic effect through the mitochondrial-dependent pathway. Taken together, our data demonstrated that *L. hatsudake* extracts exhibited an anti-cancer function via inhibiting cancer cell proliferation and promoting the apoptosis of cancer cells. This research suggests that *L. hatsudake* may be a potential anti-cancer function food. Further investigating monomers from *L. hatsudake* extract will promote our understanding of active compounds in *L. hatsudake*, which will contribute to the development of *L. hatsudake* as a novel ani-cancer function food and reduce the occurrence of cancers.

## Figures and Tables

**Figure 1 foods-11-01839-f001:**
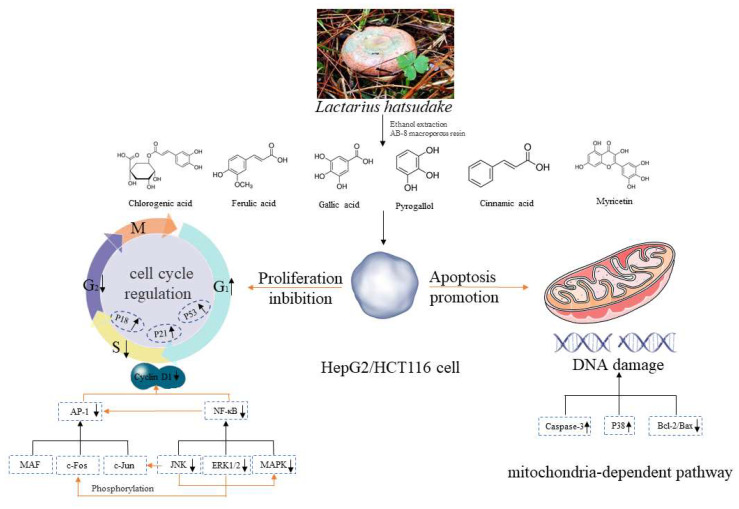
Composition and anticancer mechanism of *L. hatsudake* extracts.

**Figure 2 foods-11-01839-f002:**
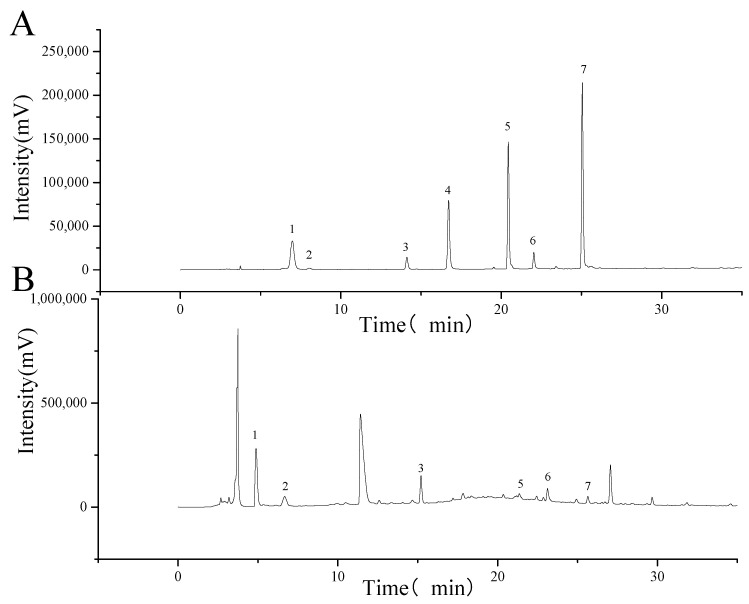
The chromatogram of *L. hatsudake* extracts analyzed by HPLC. (**A**) Standard mixture of phenolic; (**B**) phenolic profiles extracted from *L. hatsudake*. 1 gallic acid; 2 pyrogallol; 3 chlorogenic acid; 4 caffeic acid; 5 ferulic acid; 6 myricetin; and 7 cinnamic acid.

**Figure 3 foods-11-01839-f003:**
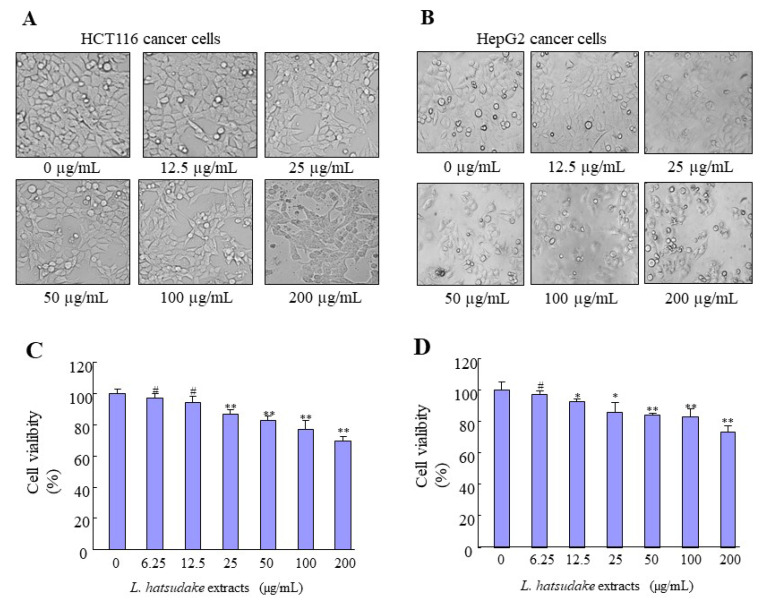
The effect of *L. hatsudake* extracts on the cell morphology proliferation of HCT116 and HepG2 cancer cells. (**A**) The cell morphology of HCT116 cells. (**B**) The cell morphology of HepG2 cells. (**C**) The cell viability of HCT116 cells by MTS analysis. (**D**) The cell viability of HepG2 cells by MTS analysis. *: *p* < 0.05; **: *p* < 0.01; #: *p* > 0.05.

**Figure 4 foods-11-01839-f004:**
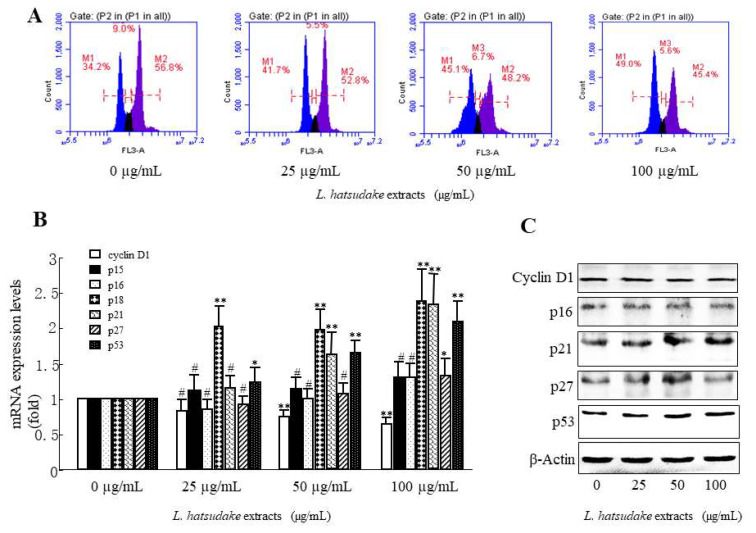
The effect of *L. hatsudake* extracts on the cell cycle, CDKIs and cyclin D1 in HCT116 cancer cells. (**A**) The effect of phenolic acids on the cell cycle of HCT116 cancer cells. (**B**) cyclin D1 and CDKI mRNA expressions; (**C**) cyclin D1 and CDKI protein expressions; *: *p* < 0.05; **: *p* < 0.01; #: *p* > 0.05.

**Figure 5 foods-11-01839-f005:**
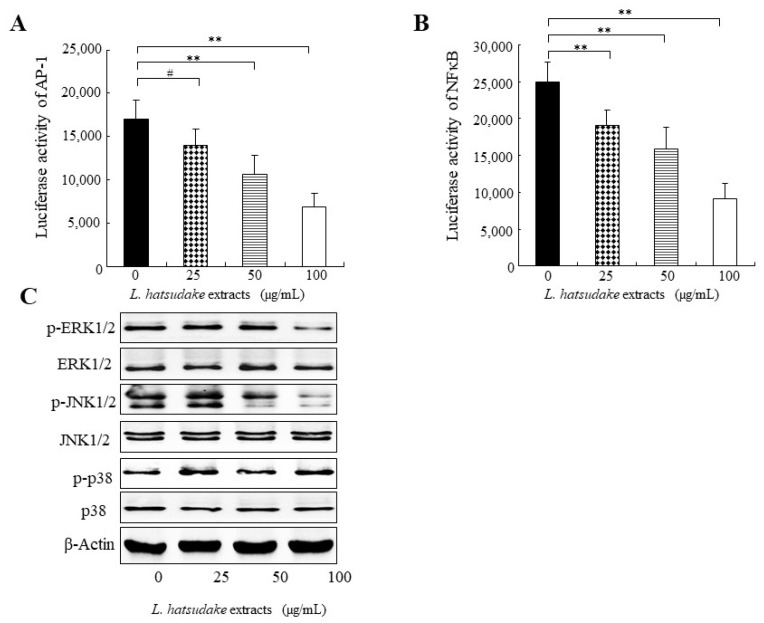
The effect of *L. hatsudake* extracts on the activities of AP-1, NFκB and MAPK signal pathway in HCT116 cancer cells. (**A**) *L. hatsudake* extracts inhibited the transcript activity of AP-1 in HCT116 cancer cells. (**B**) *L. hatsudake* extracts inhibited the transcript activity of NFκB in HCT116 cancer cells. **: *p* < 0.01; #: *p* > 0.05. (**C**) The effect of *L. hatsudake* extracts on MAPK signal pathway in HCT116 cancer cells.

**Figure 6 foods-11-01839-f006:**
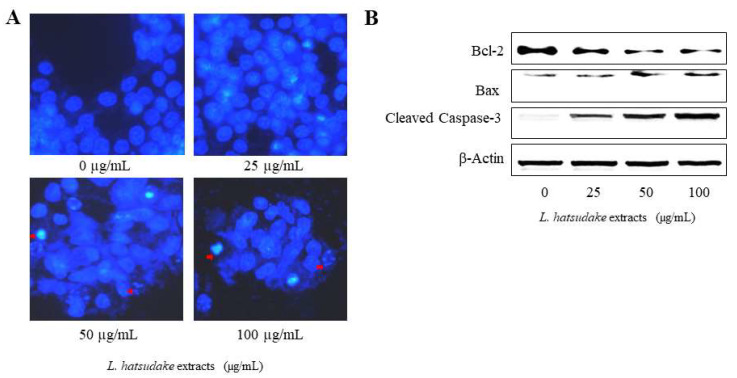
*L. hatsudake* extracts promoted HCT116 cell apoptosis and its molecular mechanism. (**A**) *L. hatsudake* extracts induced the apoptosis by Hoechst33258 fluorescent staining; (**B**) *L. hatsudake* extracts regulated apoptosis-related gene expressions by Western blotting in HCT116 cells.

**Table 1 foods-11-01839-t001:** Primers sequences used for quantitative PCR analysis of gene expression.

Primer	Forward	Reverse
p15	5′ GTT GTT TGG TTA TTG TAT GGG 3′	5′ CCC TTA TTC TCC TCA CAC AT 3′
p16	5′ CCC AAC GCA CCG AAT AGT TAC 3′	5′ GTT CTT TCA ATC GGG GAT GTC 3′
p18	5′ GGG GAC CTA GAG CAA CTT ACT AGT TT 3′	5′ AAA TCG GGA TTA GCA CCT CTA AGT A 3′
p21	5′ ATG TGG ACC TGT CAC TGT CTT GTA 3′	5′ GTT GGA GTG GTA GAA ATC TGT CAT 3′
p27	5′ AGT GGA TGA TGA GAT TGT GGA GTT 3′	5′ AAC AAG TCT AAG CTG GTG TTT TTC C 3′
p53	5′ CCC AAG CAA TGG ATG ATT TGA 3′	5′ GGC ATT CTG GGA GCT TCA TCT 3′
β-actin	5′ CAT GTA CGT TGC TAT CCA GGC 3′	5′ CTC CTT AAT GTC ACG CAC GAT 3′

**Table 2 foods-11-01839-t002:** Mass spectrometric results of substances in purified products.

	Parent Ion	Sub-Ion Ion
Chlorogenic acid	353.15	96.70
Ferulic acid	193.04	148.83
Caffeic acid	No detected	No detected
Gallic acid	187.08	124.82
Pyrogallol	125.08	97.62
Cinnamic acid	147.11	120.78
Myricetin	317.08	272.76
Substance-1	264.97	96.74, 79.72
Substance-2	187.04	124.69
Substance-3	278.77	261.09
Substance-4	309.13	96.89, 290.78, 208.70
Substance-5	433.18	152.77, 78.77, 170.47
Substance-6	295.02	276.85, 156.70, 182.72
Substance-7	149.87	95.69, 107.38, 122.78

## Data Availability

Data is contained within the article or Appendix A.

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
