# Peer review of "Phenolic Acid Profiling of Lactarius hatsudake Extracts, Anti-Cancer Function and Its Molecular Mechanisms"

_foods, 2022, doi:10.3390/foods11131839_

Round 1
Reviewer 1 Report
Dear Editor nad Authors,
The manuscript ‘Phenolic acid profiling of Lactarius hatsudake extracts, anti-cancer function and 2 its molecular mechanisms’ by Qiao Yang, Xiaoyi Zhang, Huini Qin, Feijun Luo, Jiali Ren describes research on Lactarius hatsudake extract and its composition and potential anticancer activities using Colorectal cancer cell HCT116 and hepatic cancer cell HepG2 were evaluated.
Based on the research the Authors claim that the extracts could arrest cancer cell cycle and inhibit cancer cell proliferation, which may be mediated by MAPK/NFκB/AP-1 signal pathway; the L. hatsudake extracts also promoted cancer cell apoptosis through a mitochondrial-dependent pathway.
The manuscript is generally well written. English needs some polishing. I recommend minor revision.
Some issues that I have found:
Line 25 Maybe it is worth mentioning that it is a mushroom?
Line 27 myricetin is not a phenolic acid but rather a flavonol aglycone
Line 35 as above. Maybe instead ‘six phenolic acids’ use ‘six phenolics’
Line 74 apoptosis
Line 81 myricetin?
Line 95 ultrasonic assistance or ultrasonic assisted (without ‘with’)
Line 196 use Past Simple
Line 201 as above
Line 208 as above and passive voice
‘The flask was removed and 1 ml of 1640 medium to stop digestion was added and gently pipetted to make a cell suspension’
Line 210 as above
Line 245 unknown substances appeared that need further exploration
Line 366 phenolics
Line 372 the research suggested that more attention is needed to determine the unknown substances
Yours sincerely,
Author Response
Dear editor and reviewers,
Thank you very much for your time and thoughtful comments. The grammar problems in the text have been completely modified by us according to the annotations. Below are our answers to the Reviewer’s comments. The changes were marked in red in the manuscript. We trust that all of your comments have been addressed accordingly in a revised manuscript. Thank you very much for your effort.
In the following, we give a point-by-point reply to your comments:
The reviewer recommends accepting the manuscript for publication after the manuscript is revised. The comment is as follows,
Question 1: Line 25 Maybe it is worth mentioning that it is a mushroom?
Answer: Thank you very much for your suggestion. The article has been modified as follows according to your suggestion, “Lactarius hatsudake is a mushroom with a food and medicine homology that contain numerous biologically active substances”. Please see P2 L27-28. Thank you so much!
Question 2: Line 27 myricetin is not a phenolic acid but rather a flavonol aglycone
Answer: Thank you very much for your suggestion. This article has corrected the definition of myricetin as phenolic acid, please see P2 L31-32 Thank you so much!
Question 3: Line 35 as above. Maybe instead ‘six phenolic acids’ use ‘six phenolics’
Answer: Thank you very much for your suggestion. This article has changed “Six phenolic acids” to “six phenolics”. please see P2 L31, L42, P4 L89, and P24 L465. Thank you so much!
Question 4: Line 74 apoptosis
Answer: Thank you very much for your suggestion. Thank you very much for your suggestion. This article has changed “aopotosis” to “apoptosis”. please see P4 L92. Thank you so much!
Question 5: Line 81 myricetin?
Answer: Thank you very much for your suggestion. This article has changed “myricone” to “myricetin”. please see P5 L100. Thank you so much!
Question 6: Line 95 ultrasonic assistance or ultrasonic assisted (without ‘with’)
Answer: Thank you very much for your suggestion. This article has changed “with ultrasonic assisted” to “ultrasonic assisted”. please see P5 L119. Thank you so much!
Question 7: Line 196 use Past Simple
Answer: Thank you very much for your suggestion. This article has changed “add” to “added”. please see P10 L228. Thank you so much!
Question 8: Line 201 as above
Answer: Thank you very much for your suggestion. This article has used past simple. please see P11 L240-248. Thank you so much!
Question 9: Line 208 as above and passive voice. ‘The flask was removed and 1 mL of 1640 medium to stop digestion was added and gently pipetted to make a cell suspension’. Line 210 as above
Answer: Thank you very much for your suggestion. the article has been modified as follows according to your suggestion, This article has used past simple and passive voice. please see P12 L253-265. Thank you so much!
Question 10: Line 245 unknown substances appeared that need further exploration
Answer: Thank you very much for your suggestion. These 7 substances will be further analyzed in the follow-up research. please see P14 L29-300. Thank you so much!
Question 13: Line 366 phenolics
Answer: Thank you very much for your suggestion. This article has changed “phenolic” to “phenolics”. please see P24 L463. Thank you so much!
Question 14: Line 372 the research suggested that more attention is needed to determine the unknown substances
Answer: Thank you very much for your suggestion. These 7 substances will be further analyzed in the follow-up research. please see P13 L294-295. Thank you so much!

Reviewer 2 Report
The document should be considerably improved, especially the part about results and conclusions. The format of the Journal was not applied correctly, it must be corrected.
The changes and suggestions to the document are written in the attachment

Author Response
Dear editor and reviewers,
Thank you very much for your time and thoughtful comments. The grammar problems in the text have been completely modified by us according to the annotations. Below are our answers to the Reviewer’s comments. The changes were marked in red in the manuscript. We trust that all of your comments have been addressed accordingly in a revised manuscript. Thank you very much for your effort.
In the following, we give a point-by-point reply to your comments:
The reviewer recommends accepting the manuscript for publication after the manuscript is revised. The comment is as follows,
Question 1: Line 153 how many hours? 24?
Answer: Thank you very much for your suggestion. The time was 24 h. Please see P9 L186 Thank you so much!
Question 2: L171 overnight? 8, 9, 10, 11, 12 h? L185 Time? Be specific!
Answer: Thank you very much for your suggestion. The “overnight” was 24 h and has been changed in the article. Please see P10 L209, and P11 L227. Thank you so much!
Question 3: L230 the value is not the same as the one in the table. L232 does not correspond to the value of table 1.
Answer: Thank you very much for your suggestion. The table shows the content of crude extract and polyphenols extracted from 1g dried L. hatsudake powder. What is described in this paper is the content of polyphenols in 1g crude extract and the content of polyphenols in 1g purified extract. Thank you very much!
Question 4: L231 these conditions took it from Stankunaite? Did you optimize? if so, the data is not seen.
Answer: Thank you very much for your suggestion. The optimal conditions are obtained through experiments. Please see supplementary figures 2, 3, and 4. Thank you so much!
Question 5: L233 How do you get all three values?
Answer: Thank you very much for your suggestion. The three values were through the yield of each step combined with Folin-Ciocalteau assay calculation. And considering the unnecessariness of Table 2, it has been removed. Please see supplementary figure1 and P13 L287. Thank you so much!
Question 6: L257 was its concentration determined? if they were identified, it is important that you place your concentration! especially of the chlorogenic, gallic that they mention has an important effect
Answer: Thank you very much for your suggestion. UPLC-QTOF/MS detection was used to determine the type of phenols in L. hatsudake extracts. Thank you so much!
Question 7: L266 how much do they inhibit them? it is better to put quantities
Answer: Thank you very much for your suggestion. L. hatsudake extracts inhibited cell proliferation of HCT116 with 72% cell viability and HepG2 with 75% cell viability at 200 µg/mL. Please see P16 L324-328. Thank you so much!
Question 8: L269 How many visual fields are analyzed? can be quantified by image analysis!
Answer: Thank you very much for your suggestion. we have 5 visual fields/dose, and the visual fields in this paper are generally representative images. The specific quantitative data are mainly MTS detection results.
Question 9: L275 And how is this effect compared to other mushroom or plant extracts? because it is seen that cell viability (Fig 3 C and D) remains high. You should improve your discussion with this comparison
Answer: Thank you very much for your suggestion. Compared with the phenolic extracted from Sorghum Bran had significant inhibitory at 1.5 mg/mL, the L. hatsudake extracts had inhibited cell proliferation at 25 µg/mL. Please see P16 L329-331 Thank you so much!
Question 10: L289 and L301 very poor discussion. Must improve considerably
Answer: Thank you very much for your suggestion. We rewrote the discussion part and also added new references. Please see P18 L354-358. Thank you so much!
.
Question 11: L318 and L336 This effect can be due to what compound? or is it a synergistic effect? Has this effect been found in other works in which similar compounds have been found? If we had to "blame" one of the compounds in the extract, which would it be?
Answer: Thank you very much for your suggestion. Accoding to previous investigations, chlorogenic acid, ferulic acid, caffeic acid, gallic acid, pyrogallol, cinnamic acid and myricetin all have anti-cancer effects. This effect can be due to all 6 compounds. If we had to "blame" one of the compounds in the extract, chlorogenic acid would it be “blamed”, because the content of chlorogenic acid is the highest and its anti-cancer efficiency resemble others.
Question 12: L357 It would be very good to know if the main compounds have this effect alone or if you require the small ones to produce it. If the standards were available, why were they not tested at the same concentrations?
Answer: Thank you very much for your suggestion. According to previous investigations, main compounds such as chlorogenic acid, ferulic acid, caffeic acid, gallic acid, pyrogallol, cinnamic acid, and myricetin all have anti-cancer effects. The anti-cancer effect is mainly produced by main compounds. This article is mainly to assess the anti-cancer function of L. hatsudake extracts and anti-cancer molecular mechanism, which will promote to develop the L. hatsudake as anti-cancer function food. In future research, we will continue to explore the effects alone of main compounds and standards based on this.
Question 13: L356-366 this is not conclusion. this conclusion is not adequate, because the extracts were tested, not the compounds
Answer: Thank you very much for your suggestion. We rewrote the conclusion part. Please see P19 L371-385. Thank you so much!

Reviewer 3 Report
Dears,
The authors presented a manuscript entitled "Phenolic acid profiling of Lactarius hatsudake extracts, anti-cancer function and its molecular mechanisms ".
There are small suggestions for improvement.
Please insert in abstract: Line 36 ... to apply cancer prevention, adjuvant in treatment…
Page 4: I did not find the citation of figure 1 in the text.
Results: Page 11. I consider Table 2 unnecessary, as all the information is in the text.
Add that are equivalents of gallic acid in results lines 230 to 234.
In Table 3, is the substance morricone? Please verify.
In Figure 3 C, D; Figure 4 B, C; Figure 5 A,B and Figure B: Please correct the axis legend, put it below concentrations
Conclusion: Line 367 - Please delete "higher"
Author Response
Dear editor and reviewers,
Thank you very much for your time and thoughtful comments. The grammar problems in the text have been completely modified by us according to the annotations. Below are our answers to the Reviewer’s comments. The changes were marked in red in the manuscript. We trust that all of your comments have been addressed accordingly in a revised manuscript. Thank you very much for your effort.
In the following, we give a point-by-point reply to your comments:
The reviewer recommends accepting the manuscript for publication after the manuscript is revised. The comment is as follows,
Question 1: Please insert in abstract: Line 36 ... to apply cancer prevention, adjuvant in treatment…
Answer: Thank you very much for your suggestion. The abstract has been inserted. please see P2 L43-44. Thank you so much!
Question 2: Page 4: I did not find the citation of figure 1 in the text.
Answer: Thank you very much for your suggestion. The citation of figure 1 has been added. please see P4 L93. Thank you so much!
Question 3: Results: Page 11. I consider Table 2 unnecessary, as all the information is in the text.
Answer: Thank you very much for your suggestion. Table 2 has been delated. please see P13 L287. Thank you so much!
Question 4: Add that are equivalents of gallic acid in results lines 230 to 234.
Answer: Thank you very much for your suggestion. The “equivalents of gallic acid “have been added. please see P13 L281. Thank you so much!
Question 5: In Table 3, is the substance morricone? Please verify.
Answer: Thank you very much for your suggestion. The substance morricone should be myricetin. please see P15 L314. Thank you so much!
Question 6: In Figure 3 C, D; Figure 4 B, C; Figure 5 A,B, and Figure B: Please correct the axis legend, and put it below the concentrations
Answer: Thank you very much for your suggestion. Figure 3 C, D; Figure 4 B, C; Figure 5 A,B, and Figure B has corrected the axis legend. please see P17 L337, P18 L360, and P21 L407. Thank you so much!
Question 7: Conclusion: Line 367 - Please delete "higher"
Answer: Thank you very much for your suggestion. The conclusion of "higher"has been deleted. please see P22 L465. Thank you so much!